# Sea level extremes and compounding marine heatwaves in coastal Indonesia

Weiqing Han [1] ✉, Lei Zhang [1,2], Gerald A. Meehl [3], Shoichiro Kido [4], Tomoki Tozuka [4,5], Yuanlong Li [1,6], Michael J. McPhaden [7], Aixue Hu [3], Anny Cazenave[8], Nan Rosenbloom [3], Gary Strand [3], B. Jason West[9] & Wen Xing[2]

Low-lying island nations like Indonesia are vulnerable to sea level Height EXtremes (HEXs). When compounded by marine heatwaves, HEXs have larger ecological and societal impact. Here we combine observations with model simulations, to investigate the HEXs and Compound Height-Heat Extremes (CHHEXs) along the Indian Ocean coast of Indonesia in recent decades. We find that anthropogenic sea level rise combined with decadal climate variability causes increased occurrence of HEXs during 2010–2017. Both HEXs and CHHEXs are driven by equatorial westerly and longshore northwesterly wind anomalies. For most HEXs, which occur during December-March, downwelling favorable northwest monsoon winds are enhanced but enhanced vertical mixing limits surface warming. For most CHHEXs, wind anomalies associated with a negative Indian Ocean Dipole (IOD) and co-occurring La Niña weaken the southeasterlies and cooling from coastal upwelling during May-June and November-December. Our findings emphasize the important interplay between anthropogenic warming and climate variability in affecting regional extremes.

Extreme sea level events are one of the most consequential manifestations of climate change[1,2]. Anthropogenic global sea level rise over the past century has magnified flooding and caused clear-sky floods in many coastal regions around the world[3]. While much emphasis has been placed on sea level extremes induced by storms and high tides on daily time scales[4], sea level extremes driven by climate variability and their evolution under anthropogenic climate change have received less attention. As the most dominant interannual climate mode, the El Niño - Southern Oscillation (ENSO) has global impacts on climate[5]. Over the tropical Indian Ocean, El Niño (i.e., positive phase of ENSO) often

instigates strong marine heatwaves in the Indonesian-Australian basin during boreal winter-spring[6]. The 2015–2016 El Niño initiated a strong and prolonged marine heatwave in the Indonesian-Australian basin that peaked in March 2016, and the 2016 negative Indian Ocean Dipole (IOD[7]) sustained the marine heatwave during the following boreal summer-fall[8].

While sea level Height EXtreme (HEX) events and marine heatwaves can have large ecological, economic, and social consequences individually[9], in combination they can be much more devastating, like compound extremes over land (e.g., droughts and heatwaves)[10] which

[1]Department of Atmospheric and Oceanic Sciences, University of Colorado, UCB 311, Boulder, CO 80309, USA. [2]State Key Laboratory of Tropical Oceanography, South China Sea Institute of Oceanology, Chinese Academy of Sciences, Guangzhou 510301, China. [3]Climate and Global Dynamics Laboratory, the National Center for Atmospheric Research, Boulder, CO 80301, USA. [4]Application Laboratory, Research Institute for Value-Added-Information Generation, Japan Agency for Marine-Earth Science and Technology, Kanagawa, Japan. [5]Graduate School of Science, University of Tokyo, Tokyo, Japan. [6]CAS Key Laboratory of Ocean Circulation and Waves, Institute of Oceanology, Chinese Academy of Sciences, Qingdao, China. [7]Pacific Marine Environmental Laboratory, National Oceanic and Atmospheric Administration, Seattle, WA, USA. [8]Laboratoire d'Etudes en Géophysique et Océanographie Spatiales (LEGOS), 18 Av. E. Belin, 31401 Toulouse cedex 9, France. [9]Precipitation Processing System and KBR, Inc., NASA Goddard Space Flight Center, Greenbelt, MD, USA. ✉e-mail: whan@colorado.edu

are becoming more common in a warming climate[11]. Yet, integrated studies of HEX and the compounding effect of a marine heatwave – dubbed Compound Height-Heat EXtreme (CHHEX) – are still in their infancy. A better understanding of these extremes will improve risk assessments[10,12], and investigating their interplay with anthropogenic climate change and decadal-to-interdecadal climate variability (referred to in short as 'decadal' hereafter) may help improve decadal predictions and future projections of these high-impact events.

The Indian Ocean rim region hosts one-third of the world's population, mostly from developing countries with low-lying coastal areas that are highly vulnerable to climate variability and change[13]. Located at the confluence of the tropical east Indian and west Pacific Oceans within the Indo-Pacific warm pool (Fig. 1a) and being home for diversified coral reefs, Indonesia is strongly influenced by climate variability associated with monsoons[14], IOD, and ENSO. Rapid urbanization of Java island and population growth in low-lying areas[15], together with fast sinking due to ground water extraction (e.g. Jakarta is the fastest sinking city in the world), further increase vulnerability to climate variability and change[1,3], making the problem of rising sea level particularly acute in this region. Therefore, Indonesia is an ideal testbed for understanding HEX and CHHEX events in a changing climate.

Here we combine monthly in situ and satellite observations to detect climate-driven HEX and CHHEX events around Indonesian coasts of the Indian Ocean in recent decades and to understand their causes. We primarily focus on the satellite altimetry era since 1993 when accelerated global sea level rise has been detected and attributed largely to human-induced climate change[16–18]. To put our analysis in a longer-term context, we extend our analysis to the 1960s using reanalysis data - model hindcast with assimilated observational data - and model experiments. To help understand the forcing and processes governing HEXs and CHHEXs, we carry out model experiments using the Regional Ocean Modelling System (ROMS[19]), which is an ocean general circulation model (OGCM), and the Community Earth System Model version 1 (CESM1[20]), which is a coupled global climate model. To test the model dependence of simulated signals, we perform additional experiments using an independent OGCM, the Hybrid Coordinate Ocean Model (HYCOM[21]). To further assess the roles of remote equatorial Indian Ocean wind versus local longshore wind in generating HEX and CHHEX events, we employ a Bayesian dynamical linear model[22]. Additionally, the results from large ensemble experiments of the Coupled Model Intercomparison Project phase 6 (CMIP6), which are assessed in the Intergovernmental Panel on Climate Change Sixth Assessment Report (IPCC AR6), are also analyzed to estimate the impacts of external forcing (natural plus anthropogenic) on Indonesian regional sea level change. The multi-dataset and multi-model approach is intended to identify signals that are robust to cross-dataset and cross-model differences. See the Methods section for more details.

## Results

### Detecting height extreme (HEX) & compound height-heat extreme (CHHEX) events

Satellite altimeter data from 1993–2018[23] show rapid sea level rise along the east coasts of the tropical Indian Ocean, with a rising rate of $5.12 \pm 0.17$ mm/yr near the tide gauge location on the Java coast (Fig. 1b) compared to the $3.1 \pm 0.3$ mm/yr global mean rise[16,17,24]. Accompanied with the rapid sea level rise is weak sea surface temperature (SST) warming near Java and stronger warming around the southern coast of Sumatra (Fig. 1c). Overlying the rising trend there are large year-to-year variations, as shown by the ~10yr tide gauge record at Java coast[25] and satellite altimeter data at the nearest location (Fig. 2a). The altimeter data detect fifteen HEX events during the 26 yr (1993–2018) period, defined as monthly mean sea level anomalies (SLAs) exceeding the 90th percentile, which is a commonly used threshold for defining extreme events such as marine heatwaves discussed below[26]. The tide gauge record agrees well with the altimeter data (correlation 0.99), albeit with somewhat larger amplitudes[27–30] likely because the tide gauge contains long-period tide signals but satellite altimeter data removes them[23]. It is also possible that monthly tide gauge data includes signals of storm surges, which cannot be adequately resolved by altimeter data. Additionally, satellite altimeter data have spatial averaging but tide gauge station data do not. Nonetheless, the high consistency suggests that satellite altimeter data can be used to detect HEXs in coastal Indonesia. The SLAs (with and

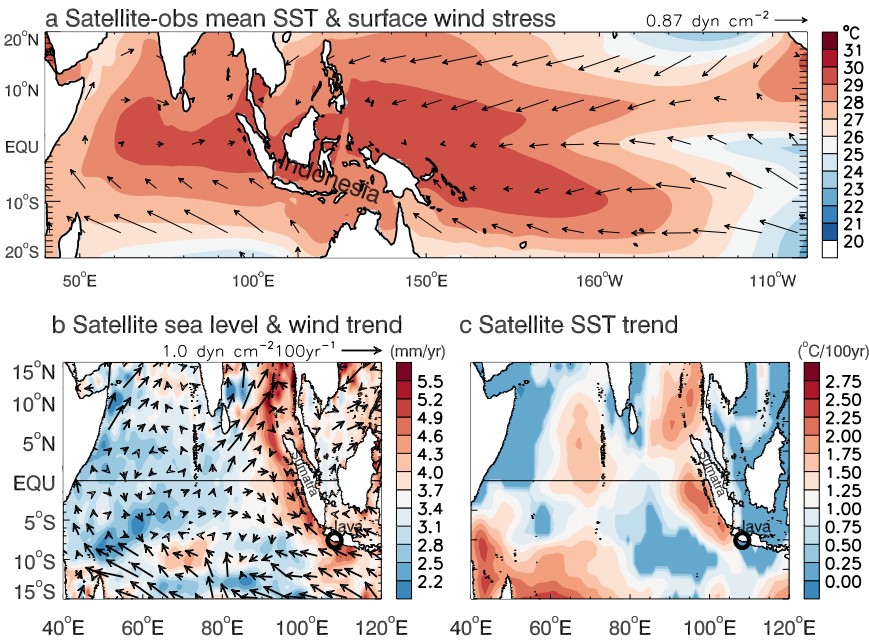

**Fig. 1 | Satellite observed sea surface temperature (SST) and surface wind stress together with trend maps of satellite sea level, surface wind, and SST. a** Mean SST and surface wind stress for the 1989–2018 period. **b** Linear trend of satellite sea level and cross-calibrated multiplatform surface wind stress from 1993–2018. **c** Linear trend of satellite SST for 1993–2018. The tide gauge location at Java coast is marked by "**o**" in **b** and **c**; its data is shown in Fig. 2a.

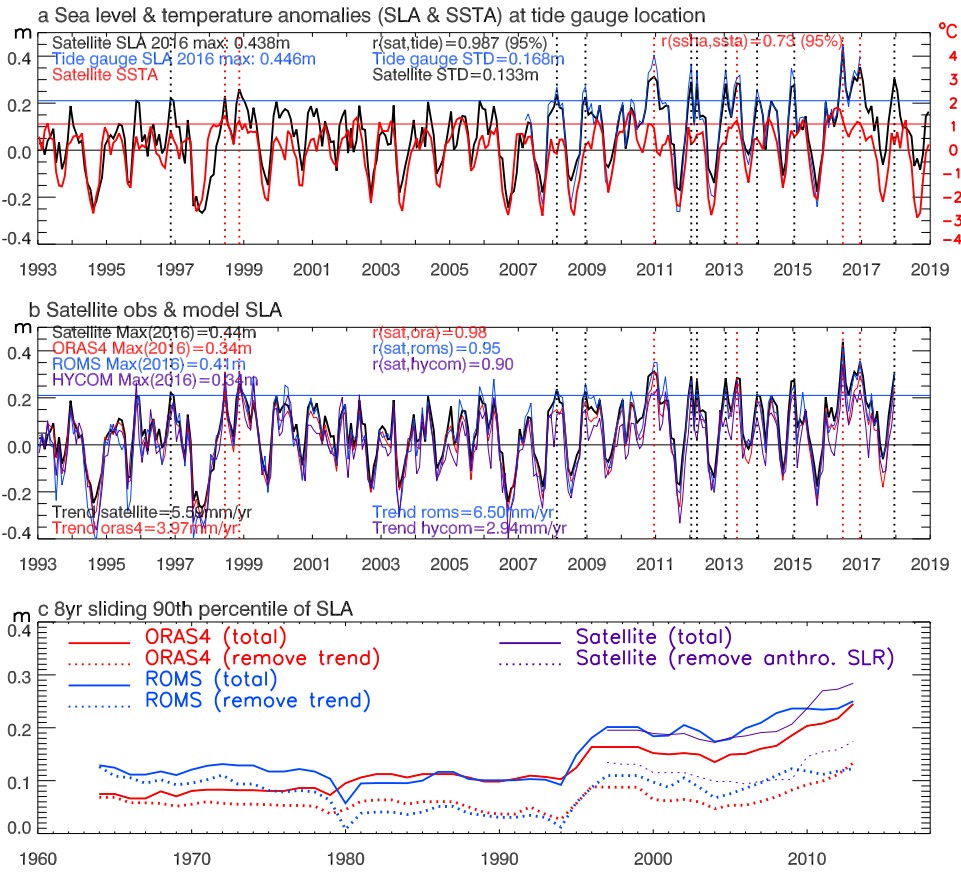

**Fig. 2 | Time series of observed and model-simulated monthly mean sea level anomaly (SLA) and sea surface temperature anomaly (SSTA) from 1993–2018 near the Cilacap B tide gauge location at Java coast (marked by "o" in Fig. 1b, c), together with 90th percentile of 8yr sliding SLA since 1960s. a** Monthly mean SLA from tide gauge during 2007–2016 (blue curve) and from the multiple-satellite-merged altimeter data at the nearest grid point (black) together with satellite observed monthly mean SSTA (red curve). The SLAs are relative to a 60yr (1958–2017) mean of ECMWF Ocean Reanalysis System 4 (ORAS4) data at the nearest location. Values exceeding the 90th percentile of altimeter data (horizontal blue line) are identified as extreme events (indicated by vertical-dotted lines) and dubbed Height EXtreme (HEX). Red dotted lines indicate HEXs co-occurred with marine heatwaves, defined as SSTA (relative to a 30yr mean from 1989–2018) exceeding 90th percentile (horizontal red line). We dub these events Compound Height-Heat EXtreme (CHHEX). **b** Monthly SLAs from satellite (black, same as that of **a**), ORAS4 reanalysis (red), and ocean general circulation model simulations from ROMS and HYCOM (blue and purple). **c** The time-evolution of 90th percentile of SLA with an 8-year sliding window from ORAS4 reanalysis (red) & ROMS simulation (blue) with & without the 1960–2017 linear trend (solid & dashed), and from satellite altimeter data (purple) with & without anthropogenic global sea level rise for 1993–2017 (solid and dashed). Note that the last value in 2013 represents the 90th percentile for 2010–2017. See Methods for data and model details.

without seasonal cycle) and the SST anomalies (SSTAs) along the entire Indonesian coasts of the south Indian Ocean (i.e., southern Sumatra, Java and Nusa Tenggara) are highly coherent, albeit with some quantitative differences (Supplementary Figs. 1b, 2), suggesting that similar, large-scale ocean dynamics control the coastal SLA and SSTA. Our discussions below primarily focus on the Java coast.

Notably, the majority (ten of fifteen) of the HEXs occur in the 8-year period of 2010–2017, with five other HEXs distributed across 1993–2009 (Fig. 2a). The strongest HEX occurs in June 2016, when monthly mean sea level rose by ~0.44 m (0.45 m) from satellite (tide gauge) observations. This monthly magnitude is comparable to the 0.5–1 m surges due to tropical storms and high tides with a return period of 100yrs along the Indonesian coasts[4,31]. The concentration of HEX events in 2010–2017 is more evident in a longer period of 1960–2017 using the European Centre for Medium-Range Weather Forecasts (ECMWF) ocean analysis/reanalysis system 4 (ORAS4) data[32] and ROMS model simulation averaged over Java coastal area (Supplementary Fig. 3, black curves). Among the fifteen HEX events, six are compounded by marine heatwaves, i.e., CHHEXs, with four CHHEXs occurring during 2010–2017 (Fig. 2a; Supplementary Table 1). Here, marine heatwaves are defined as anomalously warm water events when monthly SSTAs exceeding the 90th percentile[26]

(see Methods for details and for comparisons with heatwaves defined by daily data).

While sea level signals of the CHHEXs encompass the entire Southeast Asian coasts (Fig. 3a), the associated marine heatwaves are limited to coastal Indonesia and an area extending a few hundred kilometers offshore (Fig. 3c). By contrast, SLAs of the HEX alone events are weaker and confined to the Indonesian coasts without concurrence of marine heatwaves (Fig. 3b, d). Here, we retain the seasonal cycle when identifying HEX and CHHEX events because coastal inundation depends on full sea level magnitudes, and many marine species (e.g., corals, kelp forest) are sensitive to extreme temperature values[33,34] (see Methods). With these definitions, the extremes occur throughout the year except for July-October when coastal Indonesia is cold and sea level is low (Fig. 2a & Supplementary Fig. 1c).

## HEX concentration in 2010–2017

We hypothesize that anthropogenic global sea level rise combined with decadal increase of SLA during 2010–2017 due to natural climate variability cause the concentration of HEXs in this 8-year period. To test the hypothesis, we perform a suite of model experiments using ROMS and HYCOM. The two models and reanalysis data successfully capture the satellite observed SLAs near the Java coast (correlation

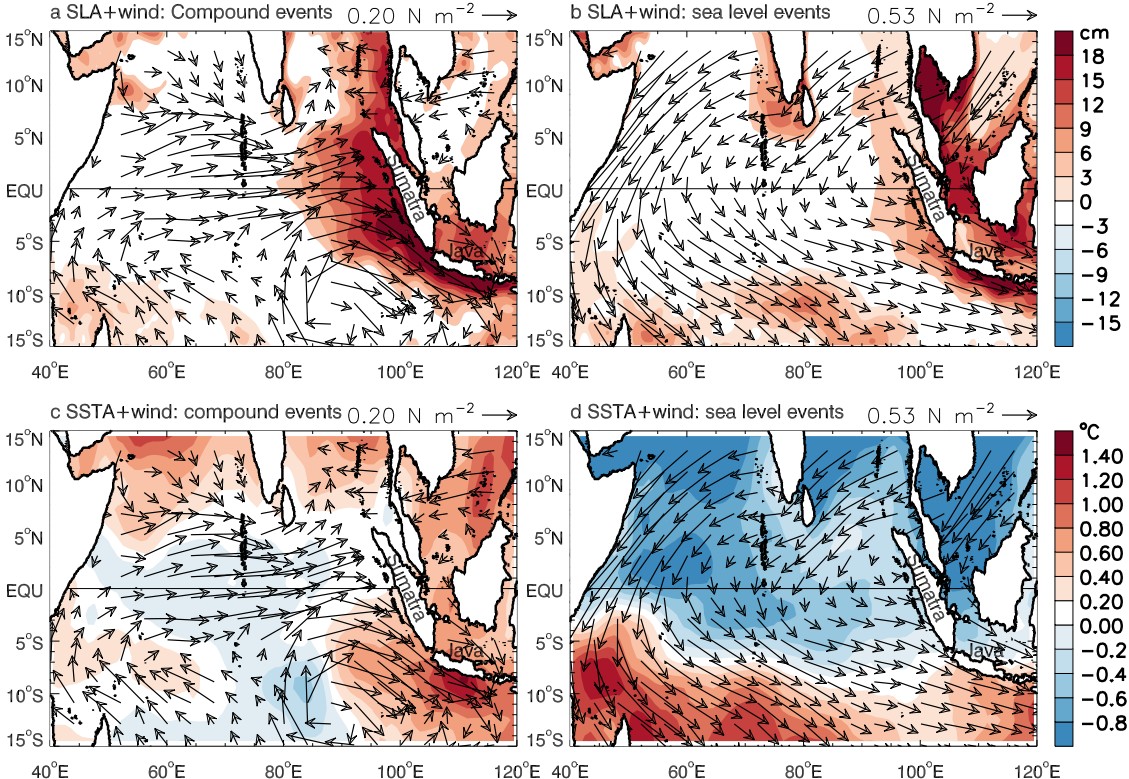

**Fig. 3 | Composite of satellite-observed monthly sea level anomaly (SLA), surface wind stress anomaly, and sea surface temperature anomaly (SSTA) for the peak months of the six CHHEX and nine HEX alone events.** All anomalies are relative to 1993–2018 mean. **a, b** Composites of SLA (color) and surface wind stress (arrows) for CHHEX & HEX alone events; **c, d** Composites of SSTA (color) and surface wind stress (arrows) for CHHEX & HEX alone events. Wind vectors are the average for the event peak month and the preceding month, considering the propagation time of equatorial Kelvin waves that impact SLA and SSTA.

0.90–0.98; Fig. 2b). HYCOM and reanalysis data, however, underestimate the satellite-observed rising trend from 1993–2017, but ROMS realistically simulates the rising trend, falling in the uncertainty range of satellite observation (Fig. 2b; Supplementary Table 2). The sea level variability magnitudes from reanalysis and models are all within data uncertainty range (Supplementary Table 2; see Methods for details). The time-evolution of HEX strength is also well simulated by ROMS compared to satellite data for their overlapping period (Fig. 2c). Both ROMS and HYCOM successfully simulate the spatial patterns and amplitudes of SLA and SSTA for CHHEX and HEX events (compare Fig. 3 and Supplementary Fig. 4). The good agreement between observations and model simulations (including ORAS4 reanalysis) suggests that the signals we identify exceed cross-model and cross-dataset differences, lending us confidence in using the models - especially ROMS - to explore the relevant forcing and processes controlling HEXs and CHHEXs.

To quantify the effects of anthropogenic sea level rise and natural decadal variability, we remove the anthropogenically-induced global sea level rise estimated from observation-based global-mean sea level dataset[18,35] (Methods) and natural decadal variability (8 yr lowpass filtered SLA) from the ROMS simulation. After removing both effects, the increased HEX occurrence and larger magnitude during 2010–2017 disappear (Fig. 4a; Supplementary Fig. 3c). The same conclusion holds after removing the linear trend and decadal variability from ORAS4 reanalysis for 1960–2017 (Supplementary Fig. 3a). By only excluding anthropogenic global sea level rise, the concentration of HEXs in 2010–2017 remains identifiable even though both frequency and magnitude are reduced (Supplementary Fig. 3, red curves of b & d). These results confirm our hypothesis that anthropogenic sea level rise combined with decadal increase of SLA during 2010–2017 – rather than randomness of HEX occurrence – causes the concentration of HEXs on the 2010–2017 period. Anthropogenic sea level rise and a

decadal increase of SLA contribute roughly equally to the enhanced HEX activities during 2010–2017 (Figs. 2c, 4b, dark red and black). Note that the effect of external forcing (natural plus anthropogenic) on dynamical sea level, which is regional sea level variation with global mean sea level rise removed, near the Indonesian coast is weak (<2 cm) with large uncertainties[36], based on the large ensemble experiments of multiple CMIP6 models (Supplementary Fig. 5).

## Causes for decadal increase of SLA in 2010–2017

The positive decadal SLA during 2010–2017, which enhances the HEXs, results mainly from surface wind stress forcing (Fig. 4b, compare black and cyan curves) associated with decadal variability of ENSO and IOD. The enhanced equatorial westerly winds over the Indian Ocean (Fig. 4d) pile up the warm pool water (Fig. 1a) in the eastern Indian Ocean and increase sea level along the Indonesian coast; meanwhile, strengthened northwesterly longshore winds near the southern Sumatra and Java coasts cause surface Ekman mass convergence toward the coasts and further enhance sea level rise there (Fig. 4d). These arguments are further supported by the Bayesian dynamic linear model forced by remote equatorial zonal wind and local longshore wind over the Indian Ocean, producing decadal SLAs similar to that of ROMS simulations (Fig. 4b, compare red, black and cyan lines).

The decadal anomalies of surface wind stress, which drive the decadal sea level increase in 2010–2017, are largely associated with ENSO decadal variability before 2012. This is because decadal SLAs along Java coast from the 10-member ensemble mean of CESM1 Pacific pacemaker experiments, which are forced by observed tropical Pacific SST (Methods), can explain a large fraction of the total and wind-driven decadal SLAs before 2012 (Fig. 4b, compare blue with black and cyan lines) and follow the decadal variability of ENSO index (blue curves in Fig. 4b, c). During the global surface warming slowdown period of ~2003–2012 when the rate of global warming decreased, ENSO decadal

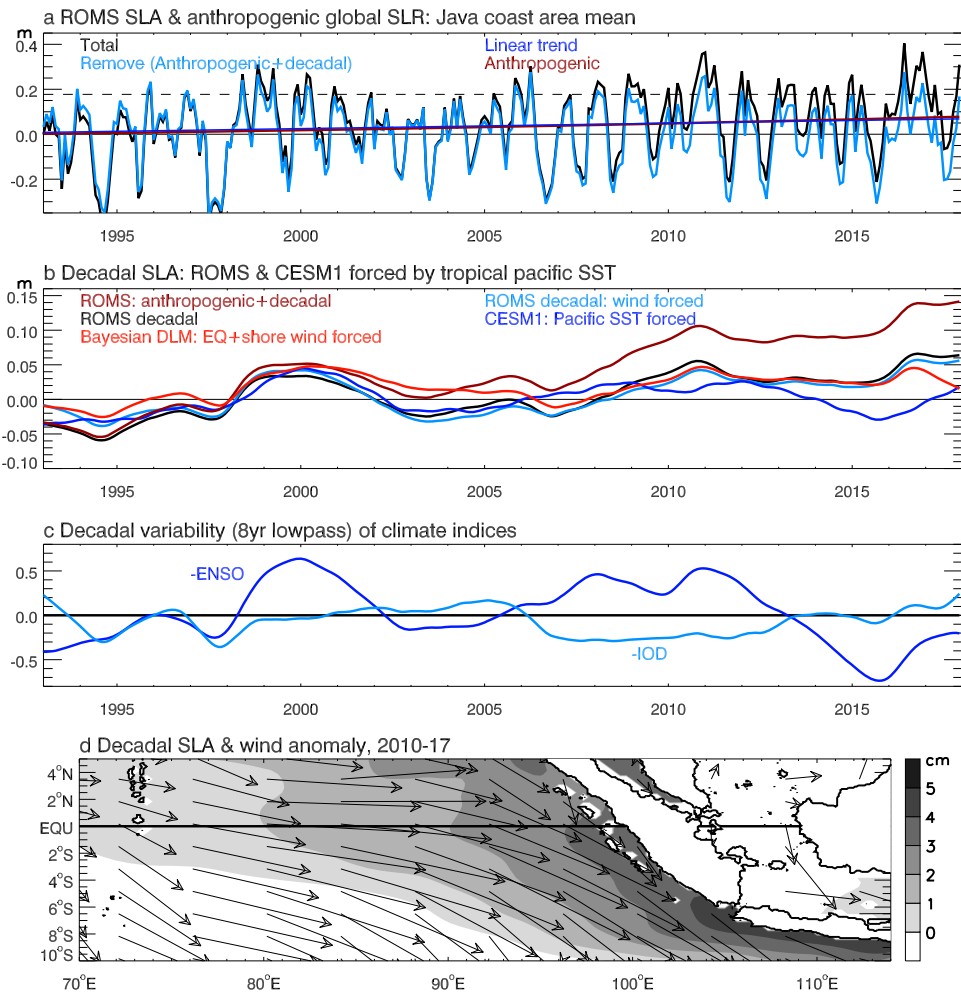

**Fig. 4 | Time series of monthly sea level anomalies (SLAs) averaged over Java coastal area (Supplementary Fig. 1) from model simulations, anthropogenically induced global mean sea level rise (SLR), climate indices, and map of sea level and surface wind anomalies averaged for 2010–2017.** Calculations are done for 1960–2017 but only 1993–2017 is shown for clarity. The 1960–2017 mean is removed from each time series. **a** ROMS simulated total SLA (black) and its linear trend (blue), observational-based estimate of anthropogenic SLR (dark red), and ROMS seasonal-to-interannual SLA with anthropogenic SLR and 8yr lowpass filtered decadal SLA removed (cyan). **b** ROMS decadal SLA (black), the sum of decadal SLA and anthropogenic SLR (dark red; which is the difference between the black and cyan curves in **a**), ROMS SLA forced only by surface wind stress (cyan), ROMS SLA from Bayesian dynamic linear model (DLM) due to equatorial zonal wind and local longshore wind forcing (red), and SLA from the 10-member ensemble mean of Pacific Pacemaker experiment using Community Earth System Model version 1 (CESM1) (blue), which assesses the impacts of tropical Pacific sea surface temperature variability. **c** Normalized indices of decadal variability (8 yr low-passed) of reversed El Niño-Southern Oscillation (-ENSO, blue) and Indian Ocean Dipole (-IOD; cyan). **d** Maps of ROMS decadal SLAs and its forcing wind stress anomalies averaged for 2010–2017.

variability is La Niña-like with intensified easterly trade winds in the tropical Pacific[37]. The intense easterly trades enhanced the mass and heat transports into the Indian Ocean from the Indonesian Through-flow (ITF)[38,39], likely also contributing to the persistent upward trend of SLA in CESM1 experiments from 2003–2009. The effects of salinity are weak in this coastal area[40]. The tropical Pacific forcing, however, cannot explain the sustained positive SLAs from 2013–2017 (Fig. 4b, blue and black). During this period, decadal variability of the Indian Ocean Dipole[7,41,42] changes from positive to negative phase, as shown by the upward trend of decadal -IOD index (Fig. 4c, cyan). Here, -IOD index is shown because negative IODs cause sea level increases along Indonesian coast. The negative IOD transition is associated with equatorial westerly and longshore northwesterly wind anomalies (Fig. 4d), which sustain the high SLAs from 2013–2017 (compare cyan curves of Fig. 4b, c).

**Individual HEX events: mechanisms**

To understand the causes for the fifteen individual SLA peaks, we analyze the seasonal-to-interannual SLA component, obtained by

removing the anthropogenic global sea level rise and 8yr-lowpass filtered decadal variability. The results show that wind stress forcing is the deterministic cause for individual HEX events (Fig. 5a, black and cyan curves). The equatorial westerly wind anomalies cause Ekman mass convergence to the equator, raising sea level. The high sea level signals propagate eastward as equatorial Kelvin waves, which subsequently propagate poleward as coastally trapped waves upon impinging on the eastern boundary, inducing coherent sea level surges along the Indonesian coasts (Figs. 3a, b; 5b). Meanwhile, the local northwesterly longshore winds induce Ekman mass convergence to the Indonesian coast, enhancing the remotely forced equatorial signals (Figs. 3a, b, 5b, red and cyan curves).

**CHHEX versus HEX-alone events**

To understand why some HEXs are accompanied by marine heatwaves (i.e., CHHEXs) while others are not, we first analyze their relationships with climate variability. Note that albeit with the strong rising trend of coastal sea level during the satellite era (Fig. 1b), the six CHHEX events remain the same after removing the 1993–2018 trends from satellite

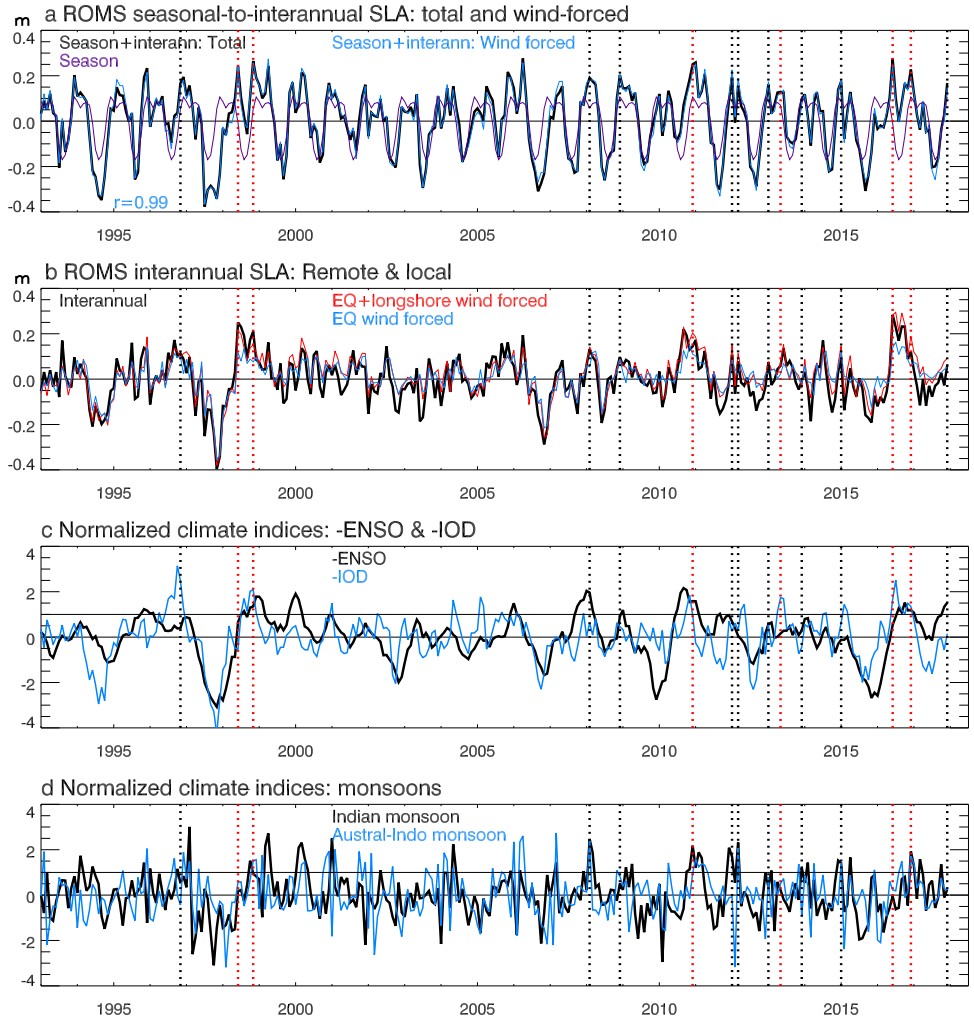

**Fig. 5 | Time series of ROMS monthly sea level anomalies (SLAs) averaged over Java coastal area and climate indices. a** Seasonal-to-interannual SLA from ROMS main run experiment (total, black) and from wind-stress forced experiment (blue), together with the mean seasonal cycle of SLA for 1960–2017 (purple). **b** Interannual SLA (seasonal cycle removed) from ROMS main run (black) and from Bayesian DLM due to remote equatorial zonal wind and local longshore wind forcing (red), and due only to remote equatorial wind forcing (blue). **c** Normalized reversed indices of seasonal-to-interannual ENSO (-ENSO; black) and IOD (-IOD; blue); La Niña and negative IOD events are identified when their indices exceed 1 standard deviation. **d** Indian monsoon wind index (black; one month lead) and Australian-Indonesian monsoon index (blue). Vertical dotted lines in each panel show the HEX (black) and CHHEX (red) events. See Methods for definition of each climate mode index.

SLA and SSTA (Supplementary Fig. 6a). For the nine HEX alone events, only the Dec 2013 HEX falls below the 90th percentile after detrending. ROMS SLAs after removing the 1993–2017 trend are close to that after removing the 1960–2017 anthropogenic global sea level rise and decadal variability ($r = 0.99$; Supplementary Fig. 6b), so the latter is used for our following discussions.

All six CHHEXs occur during negative IOD years, of which five co-occurred with La Niña (the negative phase of ENSO) although in June 2016 La Niña is developing and -ENSO index is below 1 standard deviation (Fig. 5c; Supplementary Table 1a). A negative IOD typically develops in June and peaks in September-November with warm (cold) sea surface temperature anomalies in tropical southeast (west) Indian Ocean[7,41]. An exception is 2013 when the IOD index is negative from April-October, peaks in May and becomes positive in November. The May 2013 CHHEX has no co-occurring La Niña, and its seasonal-to-interannual SLA is smaller than other CHHEXs' (Figs. 5a, b, 6c).

The negative IOD and La Niña are associated with similar patterns of surface wind anomalies in tropical Indian Ocean (Fig. 6b). Their co-occurrence intensifies the wind anomalies; by interacting with seasonal monsoon winds, they result in CHHEXs. The IOD is phase-locked with boreal summer and fall, during which seasonal southeasterly

monsoon winds prevail[43] (Supplementary Figs. 7a, b, 8). These winds cause Ekman mass divergence away from the Indonesian coast, which lowers sea level, shoals the thermocline depth (the depth range where temperature decreases the fastest towards the deeper ocean), induces seasonal upwelling of colder subsurface water to the surface, and results in a cooler SST there (Supplementary Figs. 1c, 7a, b, 8 and Fig. 6c, e). The interannual anomalies of equatorial westerly and longshore northwesterly winds associated with negative IOD and La Niña weaken or reverse the seasonally-prevailing southeasterly monsoon winds. These changed winds either reduce or reverse the seasonal coastal Ekman divergence, raise sea level, deepen the thermocline, reduce seasonal upwelling cooling and mixing of colder water from below, causing large-amplitude interannual marine heatwaves that last from June–December (dark red curves of Fig. 6c, e; Supplementary Figs. 7, 9). Meanwhile, the weakened or reversed southeasterly winds also induce anomalous southeastward longshore currents, advecting the warm equatorial water to the Indonesian coast and enhancing the warm SST anomalies.

While the interannual warm SSTAs are largely compensated by the seasonal cooling during July-October which led to weak total SSTAs (sum of seasonal and interannual SSTAs), they enhance the seasonal

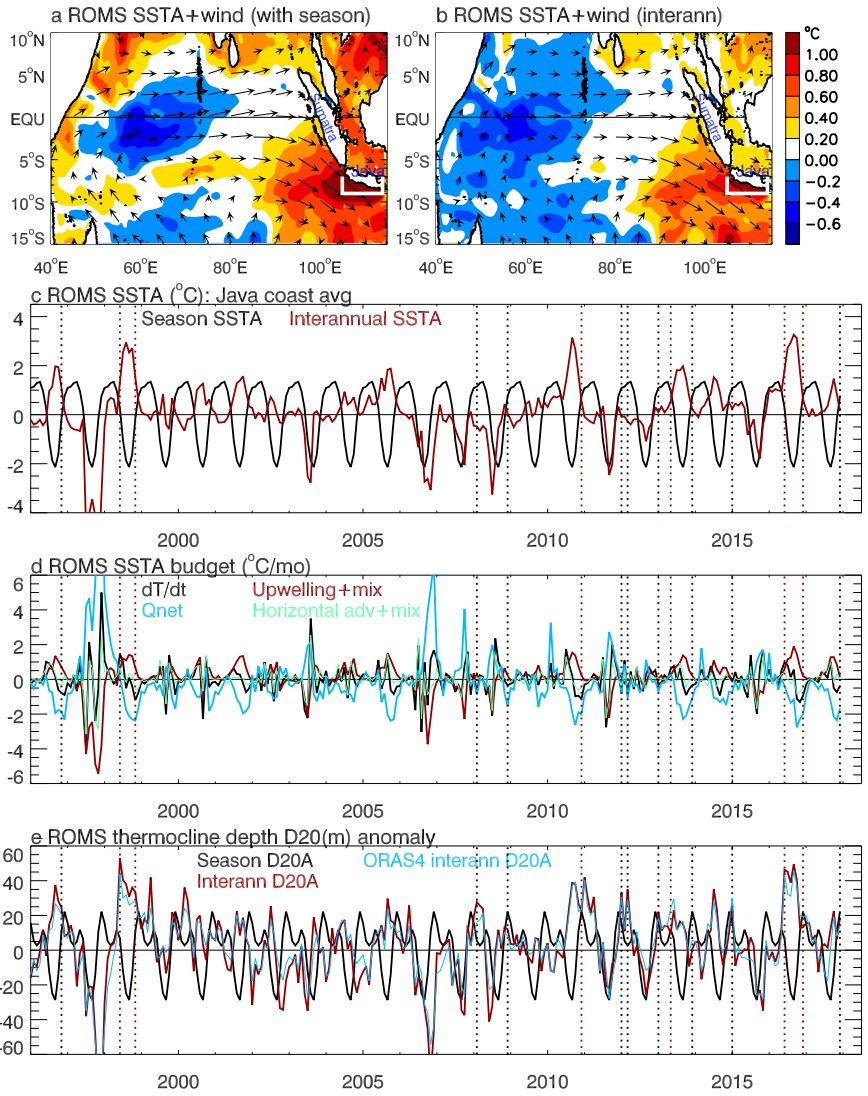

**Fig. 6 | Composites of ROMS simulated sea surface temperature anomaly (SSTA) and wind anomalies (from JRA55-do reanalysis data that force ROMS) for the six CHHEXs and time series of SSTA & its budget terms averaged in Java coastal area (white box).** **a** Composite SSTA (color) and surface wind (arrows) anomalies with the 1993–2017 mean removed but seasonal variability retained to be consistent with Fig. 3c from observations. **b** Same as **a** but with seasonal cycle removed. **c** Timeseries of mean seasonal variability (black) and interannual variability with seasonal anomaly removed (dark red). **d** Terms of heat budget analysis for mixed layer SSTA (dark red curve in **b**): time changing rate of SSTA from all processes (dT/dt, black), from net surface heat flux (cyan), from subsurface processes (upwelling+mixing, dark red) and horizontal advection+mixing (green). Units: degree per month. **e** Same as **c** except for depth of 20 °C isotherm (D20) from ROMS and ORAS4 interannual D20A, representing thermocline variability.

warm SSTAs during the IOD initiation in June and peak-to-decay period of November-December (Fig. 6c & Supplementary Fig. 9), causing the CHHEX events (Fig. 6). For the May 2013 CHHEX event, northwesterly longshore wind anomalies associated with a negative IOD work against the seasonal southeasterlies (Supplementary Fig. 8), causing a warm interannual SSTA. The moderate interannual SSTA superimposes on the high seasonal SST in May, leading to a marine heatwave (Fig. 6c).

The above arguments are further supported by the mixed layer heat budget analysis (Fig. 6d). For the 1998, 2010 and 2016 CHHEXs, reduced upwelling & vertical mixing, together with horizontal advection (Fig. 6d, dark red and green), cause the interannual warm SSTA and marine heatwaves. For the May 2013 CHHEX, increased surface heat flux together with reduced upwelling and vertical mixing accounts for the warm SSTA.

By contrast, none of the HEX alone events is associated with co-occurrence of negative IOD and La Niña, and their composite shows little interannual SSTA along the Indonesian coast (Supplementary Fig. 10). While most HEXs are associated with strong Indian and/or

Australian monsoon winds during December-March, one occurs in a negative IOD year and three occur in La Niña years (Supplementary Table 1a; Fig. 5). The Dec 2013 SLA falls below the 90th percentile with weak interannual SLA associated with monsoon variability, suggesting that with anthropogenic sea level rise and a decadal sea level increase, even weak interannual variability that occurs in the normally high sea level season can become an extreme event.

The warm interannual SSTAs associated with either a negative IOD or a La Niña are not strong enough to bring the seasonal-to-interannual SSTAs above the 90th percentile (Fig. 6c). The rest of the HEX alone events all occur during December-March; their equatorial westerly and longshore northwesterly wind anomalies associated with monsoon variability enhance the seasonal monsoons (Supplementary Figs. 8, 10), causing coastal downwelling, raising sea level and deepening thermocline. However, they increase the surface temperature very little or even slightly decrease it (Fig. 6c, d & Supplementary Fig. 10) for two reasons. Firstly, when the thermocline is already relatively deep, a further deepening does not cause a significant increase in SSTA by

reducing upwelling. Secondly, the northwesterly longshore wind anomalies enhance, rather than weaken, the seasonal monsoon winds, which strengthen the turbulent heat loss and mixing-induced cooling, counteracting the warm SSTA caused by reduced upwelling. Note that SLAs represent changes of mass and heat of the entire water column, whereas SST variability can be controlled by surface heating processes. Therefore, some marine heatwaves are not associated with sea level extremes and vice versa.

## Discussion

Satellite observations, tide gauge data, reanalysis products, and model simulations all have unique error characteristics. The fact that they are highly consistent in detecting and simulating the extreme events in coastal Indonesia demonstrates that the HEX and CHHEX events identified here well exceed data and model uncertainties. The high consistency between satellite altimetry and tide gauge observations points to the importance of continued altimetry missions and tide gauge networks in detecting and understanding sea level extremes for island nations in a changing climate. The agreement among different models on simulating the HEX and CHHEX events lends further confidence in our results. Since the 1960s, anthropogenic global sea level rise has increased the HEX magnitude near the Java coast by 0.7m–0.8 m during 2010–2017, comparable to the seasonal increase of sea level. The decadal variability of ENSO and IOD further enhance the SLAs by ~0.7 m during the 2010–2017 period, further boosting the frequency and magnitude of HEXs in the past decade. These results indicate our need for reliable decadal predictions of major climate modes, in conjunction with anthropogenic sea level rise, to achieve successful decadal predictions of regional HEX impacts.

Climate model projections suggest that continued anthropogenic warming will reduce the number of negative IOD events, which are key for generating the CHHEXs, due to a mean state change toward a shallower (deeper) thermocline in the tropical eastern (western) Indian Ocean[44-46]; however, the amplitude of the IODs is projected to increase[47]. The shallower thermocline in the eastern pole of the IOD – with continued anthropogenic sea level rise and surface warming albeit with a slower warming rate near Indonesian coast[45] – makes the upper-ocean temperature more sensitive to wind-induced Ekman convergence and thus favorably preconditions the ocean for stronger HEXs and CHHEXs in coastal Indonesia. This will increase climate change induced social, environmental, and ecological stresses.

## Methods

### Tide gauge data, satellite observations and ocean reanalysis product

The tide gauge data[25] at station Calicap B of Java coast from 2007–2016 were downloaded from the Permanent Service for Mean Sea Level (PSMSL) 2020: https://www.psmsl.org/data/obtaining/, and were corrected for Glacial Isostatic Adjustment (GIA) and Inverted Barometer (IB) effects that were provided by PSMSL along with the tide gauge data. No land movement correction was done due to the lack of GPS data within 10 km of the tide gauge station[48].

The satellite altimeter data[23] (both two-satellite and all-satellite) were download from Copernicus Climate Change Service (C3S) (2018): Sea level daily gridded data on 0.25° × 0.25° grids for the global ocean from 1993 to present, European Union, under license agreement V1.2 (Nov 2019), https://cds.climate.copernicus.eu/cdsapp#!/dataset/satellite-sea-level-global?tab=overview. Monthly means of the all-satellite data are used in our analysis, and the timeseries shown in Fig. 2 is from the nearest grid point approximately 18 km southeast of the Java tide gauge station. Using the two-satellite data yields similar results except for slightly weaker amplitudes for some extreme events.

The Cross-Calibrated Multi-Platform (CCMP) Satellite derived winds[49,50] were downloaded from http://www.remss.com/measurements/ccmp/. The National Oceanic and Atmospheric

Administration (NOAA) blended satellite sea surface temperature (SST) data[51] on 1° × 1° grids at monthly resolution and on 0.25° × 0.25° at daily resolution are publicly available at: (https://psl.noaa.gov/data/gridded/data.noaa.oisst.v2.html; https://psl.noaa.gov/data/gridded/data.noaa.oisst.v2.highres.html).

The European Centre for Medium-Range Weather Forecasts (ECMWF) operational ocean analysis/reanalysis system version 4 (ORAS4)[32] monthly sea level and temperature data at 1° × 1° resolution, which are used to infer thermocline depth (as indicated by the depth of 20 °C isotherm), from 1958–2017 are obtained from https://www.ecmwf.int/en/research/climate-reanalysis/ocean-reanalysis. The ORAS4 data are ocean model hindcasts assimilated observational data, including satellite altimeter data.

### Estimates of anthropogenic global sea level rise

First, we obtained the monthly global mean sea level (GMSL) data from CSIRO available for 1880–2013, which are adjusted to satellite observations from 1993–2013[18] (ftp://ftp.csiro.au/legresy/gmsl_files). Then we use the 1880–1992 GMSL from this dataset and the NASA monthly GMSL data from 1993–2019[24] to form a time series from 1880–2019, and choose the 1960–2019 period for our analysis. The NASA GMSL data are downloaded from http://podaac.jpl.nasa.gov/dataset/MERGED_TP_J1_OSTM_OST_ALL_V42[52]. Note that the CSIRO and NASA GMSL data are very similar for their overlapping period of 1993–2013. Two methods were used to assess the anthropogenic GMSL rise (GMSLR): (1) Since anthropogenic effect (thermal expansion, land ice melting and land water storage) explains ~90% of the GMSL in recent decades[35,53], we use 90% of the quadratic fits of GMSL (i.e., fitted GMSLR*0.9) to represent anthropogenic GMSLR; the quadratic fits are done individually for the 1960–1992 and 1993–2019 periods to consider SLR acceleration in recent decades; (2) For the 1993–2019 satellite period, we use the climate-change induced acceleration of 0.084 mm yr$^{-2}$ [17] to estimate the anthropogenic GMSLR, and keep the 1960–1992 period the same as in (1). The two curves are almost identical.

### CMIP6 climate model simulations

The coupled model intercomparison project phase 6 (CMIP6) large ensemble experiment results, with ensemble members of each model ranging from 10–50 (Supplementary Fig. 5), were obtained from https://esgf-node.llnl.gov/projects/cmip6/. They are used to assess the impacts of external forcing (natural + anthropogenic) on regional sea level near the Indonesian coast.

### Climate mode indices

The monthly HadISST data available since 1870[54] are used to calculate climate mode indices. The climatological seasonal cycle is removed before we calculate the indices. Climate events are defined as indices exceeding one standard deviation. The Niño3.4 index, which is the time-series of SST anomaly (SSTA) averaged for (120°W-170°W, 5°S-5°N), is used to represent ENSO. ENSO is the most dominant mode of climate variability, which is associated with strong SSTA in the tropical Pacific Ocean and has large impacts on global climate. It develops during boreal summer and peaks during boreal winter (Dec-Feb). Its negative (cold) phase is referred to as La Niña, and positive (warm) phase is called El Niño. The decadal variability of Niño3.4 index, obtained by 8 yr lowpass filtering, represents decadal variability of ENSO, which is highly correlated with the Interdecadal Pacific Oscillation (IPO)[55], with its negative phase being referred to as La Niña-like and positive (warm) phase being El Niño-like SSTA pattern.

The dipole mode index, defined as the SSTA difference between tropical western Indian Ocean (50°E-70°E, 10°S-10°N) and tropical eastern Indian Ocean (90°E–110°E, 0°–10°S), represents the Indian Ocean Dipole (IOD)[7]. In general, the IOD develops in boreal summer and peaks during boreal fall (Sep-Nov). Its negative phase is associated

with warm SSTA and deeper thermocline in the eastern pole and cold SSTA and shallower thermocline in the western pole.

The monthly wind shear index[56] is used to represent Indian monsoon variability, which is the zonal wind U at 850hPa (U850) averaged over (40°E-110°E, EQ-20°N) minus that of 200hPa (U200), i.e. U850(40-110E,EQ-20N)-U200(40-110E,EQ-20N), and it affects equatorial zonal wind anomaly (Supplementary Fig. 11a). The Australian-Indonesian monsoon index[43] is defined as U850 anomaly averaged over (110°E-130°E, 15°S-5°S), which affects longshore wind anomaly (Supplementary Fig. 11b). Both are calculated from NCEP1 reanalysis winds from https://psl.noaa.gov/data/gridded/data.ncep.reanalysis.html[57].

## Definitions of marine heatwave (MHW) and Compound Height-Heat EXtreme (CHHEX)

The NOAA blended satellite SST data[51] are used to detect marine heatwaves (MHWs). The MHWs are defined as monthly SST anomalies relative to the mean of a 30yr baseline period of 1989–2018 exceeding the 90th percentile, following the recommended definition of MHWs from the previous studies[26,58,59]. Based on this definition, the mean seasonal variation of SST is retained when we define MHW events because marine ecosystems are sensitive to the total SST magnitude, although interannual variability of SST excluding the mean seasonal cycle is also meaningful for some species[26,58,59]. Note that there are previous studies using monthly SST to define MHWs[26]. Although a general recommendation on MHW definition has been given, the choice of threshold and calculation of SST anomalies should be based on the study purpose. Comparing to the MHWs identified using daily data, which are defined as discrete prolonged anomalously warm water events when daily SSTAs exceed the 90th percentile for the 30 yr baseline period of 1989–2018 and persist for at least 5 days[26], we see that most MHWs identified by monthly SST data correspond to a series of MHWs defined by daily SST data (Supplementary Fig. 12), except for Nov 1998 and Dec 2016. The stronger and longer-lasting MHWs based on monthly data correspond to a series of more intense and/or more frequent MHWs from daily data (Supplementary Fig. 12).

Using monthly data, a CHHEX event is identified when a MHW (i.e., monthly SSTA > 90th percentile) is detected during a HEX event. Note that for the December 2010 event, SSTA merely reaches the 90th percentile two months before the HEX peak but remains close to the 90th percentile when HEX peaks. Thus, we also count this event as a CHHEX. A gap of at least one month is required between two consecutive HEX (or MHW) events.

## Ocean general circulation models (OGCMs), experiments and validation

To ensure the HEX and CHHEX events detected here exceed cross-model differences, we use two independent OGCMs with somewhat different surface forcing fields to carry out experiments: The Regional Ocean Modeling System (ROMS[19]) and the HYbrid Coordinate Ocean Model (HYCOM[21]). The ROMS is configured for the global tropical oceans (25°S to 25°N) with a horizontal resolution of 1/3° × 1/3° and 40 vertical sigma layers[60], and forced by 3hourly Japanese 55-year atmospheric reanalysis - drive ocean (JRA55-do[61]) fields (e.g., surface wind, heat flux and precipitation) from 1958–2017, which are the JRA55 reanalysis surface fields adjusted relative to reference datasets. Along the northern and southern open ocean boundaries, the mixed radiation-nudging boundary condition is used, where temperature, salinity, and horizontal velocity are relaxed to the monthly values of ORAS4 reanalysis data with the nudging time scale of 360 days (3 days) for the outflow (inflow) case. The open ocean boundary conditions allow the influence of global sea level rise on Indonesian coast because ORAS4 reanalysis assimilated observed data (including satellite altimeter data), and there is no constraint for volume conservation over a specific ocean basin.

Two experiments were performed for the 1958–2017 period: *ROMS main run (MR) & ROMS WSTRESS* run. The MR is the complete

solution, and the WSTRESS run is the same as the MR except for fixing the forcing fields used to calculate heat and freshwater fluxes to their climatology but keeping 3hourly wind stress forcing as in the MR. Therefore, ROMS WSTRESS run isolates oceanic variability driven only by surface wind stress.

A recent version of HYCOM was set up for the global ocean with 50 hybrid layers, 1/2° × 1/2° resolution, and daily surface forcing fields from JRA55 reanalysis dataset from 1958–2017. Note that global sea level rise due to land ice melting, which contributes ~44% during the satellite altimetry era[16], is not included in the model.

Overall, the reanalysis data and model simulations successfully capture the satellite observed SLAs near the Java coast, with correlation with satellite SLA being 0.98 for ORAS4 reanalysis, 0.95 for ROMS and 0.90 for HYCOM (Fig. 2b; Supplementary Table 2). The linear trend of ROMS main run SLA is 6.50 ± 1.16 mm/yr, which is within the uncertainty range of satellite SLA trend of 5.59 ± 0.99 mm/yr for the 1993–2017 period. The ORAS4 reanalysis data – which assimilate satellite SLA – underestimates the sea level rise trend, as does HYCOM, with both exceeding the uncertainty range of satellite data. This is likely due to the coarser 1° × 1° resolution of ORAS4 reanalysis data with the nearest grid point being farther away from the tide gauge location compared to the 0.25° × 0.25° satellite observation. The global HYCOM significantly underestimates the sea level rising trend along the Indonesian coast, in part due to the missing land ice melting effect in the model. The underestimation of the sea level rise trend in HYCOM without including land ice melting, and the adequate simulation of sea level rise trend in ROMS that includes the effect of land ice melting by using ORAS4 reanalysis data as boundary conditions, further confirm the impact of global sea level rise on Indonesian coastal sea level change.

Despite errors in simulating the sea level rise trend in HYCOM and ORAS4 reanalysis, the increased occurrence of HEX events during 2010–2017 is consistent in all datasets. Since ROMS applies open boundary conditions with 3hourly forcing fields, it contains global sea level rise and storm surge signals like the tide gauge data. This is probably why the ROMS SLAs are somewhat larger than satellite data, as the tide gauge observation (Fig. 2a, b). Due to the stronger amplitudes, more HEXs are identified in the tide gauge record and the ROMS simulation based on the 90th percentile threshold of satellite data. Since this study aims for climate-driven longer timescale extremes, we focus on the events identified using monthly satellite altimeter data.

After removing the 1993–2017 trend, the standard deviation of satellite SLA is 0.12 m, compared to the 0.13 m in ORAS4, 0.15 m ROMS, and 0.13 m in HYCOM. All of them are within the 0.04 m difference between tide gauge and satellite data (Supplementary Table 2), suggesting that the sea level variability magnitudes in both reanalysis data and model simulations fall in the uncertainty range of observations.

The time-evolution of HEX strength, represented by the 90th percentile of SLAs with an 8year sliding window, is well simulated by ROMS compared to satellite data for their overlapping period (Fig. 2c, solid blue and purple curves). In comparison, the ORAS4 reanalysis data underestimate the HEX magnitude during the satellite era (Fig. 2c, solid red), likely due to its underestimation of the rising trend. The spatial patterns and amplitudes of SLA and SSTA associated with the CHHEX and HEX events from ROMS and HYCOM (Supplementary Fig. 4) agree well with those of satellite observations (Fig. 3). The good agreement between observations and model simulations (including ORAS4 reanalysis) suggests that the signals we identify exceed cross-model and cross-dataset differences, which give us confidence in using the models - especially the ROMS - to explore the relevant forcing and processes controlling the HEXs and CHHEXs.

## Coupled global climate model experiments using CESM1

To assess the role played by ENSO and its decadal variability in affecting Indian Ocean sea level, we perform a ten-member ensemble of the

tropical Pacific Ocean pacemaker experiments using the National Center for Atmospheric Research (NCAR) Community Earth System Model version 1 (CESM1[20]) from 1920–2019. In this experiment ensemble, SST in the central and eastern tropical Pacific is restored to observations but is fully coupled to the atmosphere elsewhere. The 10-member ensemble mean fields of the pacemaker experiments estimate the Pacific impacts on the Indian Ocean through both atmospheric bridge and oceanic connection via the Indonesian Throughflow. Even though the model has some biases[62], its results provide valuable assessments of remote forcing from the Pacific especially in the context of analyzing these results with observations and standalone OGCM simulations.

### ROMS mixed layer heat budget analysis

Time evolution of the mixed layer temperature, $T_{mix}$, is governed by the following equation:

$$
\frac{\partial T_{mix}}{\partial t} = \underbrace{\frac{Q_{net}}{\rho C_p h} - \frac{Q_{sw}(z=-h)}{\rho C_p h}}_{\text{Surface heat flux}}
$$
$$
\underbrace{-\frac{1}{h}\int_{-h}^{0}\left(u\frac{\partial T}{\partial x}\right)dz - \frac{1}{h}\int_{-h}^{0}\left(v\frac{\partial T}{\partial y}\right)dz + \frac{1}{h}\int_{-h}^{0}\nabla_h\cdot(\kappa_h\nabla_h T)dz}_{\text{horizontal advection \& mixing}} \quad (1)
$$
$$
\underbrace{-\frac{1}{h}\int_{-h}^{0}\left(w\frac{\partial T}{\partial z}\right)dz - \frac{1}{h}\left(\kappa_v\frac{\partial T}{\partial z}\right)_{z=-h} - \frac{\Delta T}{h}\frac{\partial h}{\partial t}}_{\text{Subsurface process}}.
$$

where $T$ is the sea water temperature, $\rho$ represents the sea water density, $C_p$ is the specific heat of the sea water, ($u$, $v$, $w$) denote zonal, meridional and vertical velocity, respectively, and $h$ is the mixed layer depth. The mixed layer depth $h$ is defined as a depth at which the potential density increases by 0.01 kg/m³ from the sea surface. $Q_{net}$ is the net surface heat flux and $Q_{sw}$ ($z = -h$) is the shortwave radiation at the bottom of the mixed layer. Additionally, $\kappa_H$ and $\kappa_v$ are horizontal and vertical mixing coefficients, and $\Delta T$ is the temperature difference between the mixed layer and upper thermocline. The first two terms on the right-hand side represent the surface heat flux forcing; the third-to-fifth terms are zonal advection, meridional advection, and horizontal mixing. The last three terms represent subsurface processes: vertical advection, vertical mixing, and entrainment, respectively. The mixed layer heat budget is closed in the ROMS experiment[60,63].

### The Bayesian dynamical linear model

To quantify forcing by remote equatorial wind and local longshore wind on sea level variability along the Indonesian coast, we apply the Bayesian dynamic linear model (DLM) with two predictors. The Bayesian DLM consists of two equations: an "observation equation" analogous to the conventional multiple linear regression model (Eq. (2) below), and a "state equation" that controls the dynamical evolution of coefficients $b_i$ ($i = 0, 1, 2$) represented by Eq. (3).

$$
Y(t) = b_0(t) + b_1(t)X_1(t) + b_2(t)X_2(t) + \varepsilon(t), \varepsilon(t) \sim N(0,V(t)), \quad (2)
$$

$$
b_i(t) = b_i(t-1) + w_i(t), w_i(t) \sim N(0,W_i(t)). \quad (3)
$$

In equation (2), $X_1$ and $X_2$ are the predictors, and $Y(t)$ is the predictand. The state Eq. (3) means that the predictive distribution of $b_i$ at each time step $t$ (i.e., *posterior*) is updated based on its previous step $t-1$ distribution (i.e., *prior*) and the probability of observations $Y$ conditional on $b_i$ at time $t$ (i.e., the *likelihood*) using Bayes theorem[22]. Coefficients $b_i$ are obtained by applying Kalman filtering and smoothing, with the regression coefficient of conventional linear regression as its initial guess[64,65]. The $b_0(t)$ term represents a time-varying "intercept" whose variability is unexplained by the predictors $X_i$, while the $b_i$ terms represent the non-stationary influence of $X_i$ on $Y$, which is superior to

the conventional regression model with stationary $b_i$ which can only estimate stationary impacts of the predictors[64]. Terms $\varepsilon(t)$ and $w_i(t)$ are independent white noise or errors, distributed normally with a mean of 0 and variances of $V(t)$ and $W_i(t)$. Here, we use zonal wind stress anomalies averaged over the equatorial area (65°E-95°E, 5°S-5°N) and longshore wind stress averaged along Sumatra and Java coast (Supplementary Fig. 1) as the two predictors ($X_1$ and $X_2$) and sea level anomalies along Indonesian coast as the predictand, $Y(t)$. Time series of the equatorial wind ($X_1$) leads Java coast sea level anomaly by one month to consider the propagation time of equatorial Kelvin wave, but the local longshore wind has no lag.

### Data availability

All the observational data sets used in this research are publicly available from links provided in the Methods section. The model data generated in this study, including the OGCM experiments using ROMS and HYCOM, CESM1 Pacific Pacemaker experiments and the Bayesian dynamic linear model that were used to produce the Figures in the main text (Figs. 1–6) have been deposited at the University of Colorado Scholar database (https://doi.org/10.25810/mzt8-w960).

### Code availability

The IDL and MATLAB codes for carrying out the analyses and producing the figures are deposited at a public repository at the University of Colorado Scholar (https://doi.org/10.25810/mzt8-w960).

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

## Acknowledgements

W.H. and L.Z. are supported by NASA Ocean Surface Topography Science Team award 80NSSC21K1190 and National Science Foundation award NSF-AGS 1935279. Y.L. is supported by the Strategic Priority Research Program of Chinese Academy of Sciences through grant XDB42000000. GAM, AH, NR and GS are supported by the Regional and Global Model Analysis (RGMA) component of the Earth and Environmental System Modeling Program of the U.S. Department of Energy's Office of Biological & Environmental Research (BER) via National Science Foundation IA 1947282. W.X. is supported by Research Project Program of State Key Laboratory of Tropical Oceanography LTORC2202. This work also was supported by the National Center for Atmospheric Research, which is a major facility sponsored by the National Science Foundation (NSF) under Cooperative Agreement No. 1852977. The ROMS simulation is supported by the Cooperative Research Activities of Collaborative Use of Computing Facility of the Atmosphere and Ocean Research Institute, the University of Tokyo. MJM is supported by NOAA. PMEL contribution no. 5220. The CESM1 Pacemaker experiments supercomputing resources were provided by NSF/CISL/Cheyenne. We thank Qing Hu of Villanova University for carefully reading the MS and providing helpful comments. We would like to acknowledge high-performance computing support from Cheyenne (https://doi.org/10.5065/D6RX99HX) provided by NCAR's Computational and Information Systems Laboratory, sponsored by the National Science Foundation.

## Author contributions

W.H. led the project and did the main analyses and writing, L.Z. analyzed CMIP6 model results and carried out the CESM1 extension experiment from 2013 to 2019, G.A.M., A.H., N.R., and G.S. carried out the CESM1 experiments from 1920–2013, helped set up the CESM1 extension experiments and did the post-processing, S.K., and T.T. performed the ROMS experiments and provided the mixed layer heat budget analysis results, M.J.M. contributed to the scientific results through stimulating discussions and analysis, A.C. contributed to the analysis and discussion of satellite altimeter data, and B.J.W. helped to confirm the effects of atmospheric intraseasonal oscillations, although this part is not included in the revised MS. W.X. helps with the Bayesian Dynamic Linear Model experiments. W.H., L.Z., G.A.M., S.K., T.T., Y.L., M.J.M., A.H., A.C., N.R., G.S., B.J.W., and W.X. contributed to reading/writing the paper.

## Competing interests

The authors declare no competing interests.
