## [Peer Review File · Nature Communications]

Sea level extremes and compounding marine heatwaves in coastal IndonesiaREVIEWER COMMENTS

Reviewer #1 (Remarks to the Author):

Climate-driven sea level extremes and compounding effects of marine heatwaves in coastal Indonesia by Han et al.

The manuscript addresses a very important science issue on the sea level extremes and marine heat waves that are lasting for many days.

They have provided the evidences for the extreme sea levels and marine heat waves along the coast of Java.

They also highlighted the significance of the topic considering the current sinking of Java island. However, the authors could not provide any convincing hypothesis or mechanisms causing the sea level extremes and marine heat waves over the study region. The use of different models is not properly justified, and the detrending also to be relooked at before making the conclusions. The role of decadal variability is attempted but no clear conclusion is made in the manuscript on that.

The authors suggest negative Indian Ocean Dipole as the prime reason for the sea level extremes and marine heat waves over the Java coast, but the convincing hypothesis or mechanisms are not shown.

The different ocean models used in the study are forced by different flux products, making any conclusions (made out of them) very difficult.

Use of closed boundary conditions in one set of models and open boundary conditions in another set of models should also be avoided. Such experiments should be carried out with the same model.

They need to validate the models very carefully before making the conclusions.

So overall major revision is mandatory by making a clear hypothesis and proving it instead of just speculating many things.

If the authors carefully rewrite the manuscript carefully with more convincing experimental evidences, then the manuscript can become a very useful article.

Reviewer #2 (Remarks to the Author):

This is a very interesting and highly original paper on a new combination of extremes with potentially strong impacts, sea level height extremes and marine heatwaves in the Indian Ocean. The paper is well written and the authors present a thorough analysis, combining a suite of different datasets which help provide plausible hypotheses of the drivers behind those compound events. As such, the paper is an important contribution to the emerging field of compound event research.

On concern would be that the main analysis focuses on compound events in two variables of which one has a strong positive trend (sea level). This is also used to identify drivers of such events. However, due to the strong trend, the strongest driver is anthropogenic climate change, which potentially confounds more detailed conclusions about other important processes. The authors also present some results on the detrended data and I would propose to focus the entire discussion of compound event drivers more on the detrended data to more clearly separate anthropogenic and non-anthropogenic drivers. For instance, I would be interested in the co-occurrence rate of height extremes and marine heatwaves in the detrended time series and the composites of those events, which should be much more representative of short-term processes. In particular, given the very strong correlation between SSHA and SSTA (Fig 2) I would expect a relatively high compound event rate (much more than the 5 event used in the current version).

Along these lines, that the compound events only occur in the last few years is somewhat obvious if the underlying variables have such strong trends. In this context it is strange to see this phrased as a major result, e.g. by using the word "striking".

As I'm not an expert on the oceanic processes, I can merely judge the statistical analyses, which are sound and reveal interesting insights about the processes behind compound events. In particular the employment of multiple different datasets shows robustness of the results and the usage of composites is a good way to reveal important compound event drivers.

Further comments:

- L28: These numbers strongly depend on how extremes are defined. I would use more qualitative words here, e.g. "the majority" or something similar**
- Examples in the introduction are on eastern Pacific but study regions is the Indian Ocean. Maybe chose more relevant examples for the study region**
- L42: Calling extremes on a monthly resolution longer-lasting extremes is a bit of a misnomer. One can call something long-lasting if there are many subsequent extremes in a high-resolution dataset but not if the original temporal resolution is coarse. With the same logic one could call a warm year an extremely long-lasting heatwave, which is obviously nonsense.**
- L112: This is to be expected (and not "striking") if extremes are analysed in a variable with strong upward trend, as stated higher up**
- L194 and below: I would strongly suggest to present and discuss the drivers of extreme events based on the detrended time series. Otherwise the process analysis is confounded by long-term anthropogenic trends**
- Cyan color in Fig. 2 and other figures not legible, please consider using a different color**

Reviewer #3 (Remarks to the Author):

This study use historical observational data and ocean models to describe and investigate the sea level extremes off the Indonesian coast in the southeast Indian Ocean, as well as their compound events with marine heatwaves. It is found that negative Indian Ocean Dipole like condition is the key driver for the development of the compound events, whereas the sea level extreme only events are more driven by regional wind anomalies. Compound sea level and marine heatwaves events are rarely studied so that this research could contribute to the literatures to inspire new research on compound marine extremes. It is a significant amount of research and manuscript is organized with a good structure. Still, the manuscript will need to address the following comments before it can be accepted for publication.

The definition of marine heatwaves: Hobday et al. (2016) used daily data to define the marine heatwaves, with a 90th percentile criteria. In this study, the sea level extreme is defined as two standard deviations and marine heatwave is defined as one standard deviation, using monthly data. So there need some consistencies. 90th percentile is equivalent to 1.6 standard deviations for normal distributions.

As noted in various studies, there is a positive skewness in the IOD (Ogata et al. 2013; Cai et al. 2009). So for the eastern IOD pole SST, there could be a negative skewness. How would this SST skewness affect the selection of the marine heatwave events?

It is useful to compare marine heatwave selection with daily SST data, which is available from various satellite products. Monthly warming may not reflect the full impacts of marine heatwaves on biology processes.

In some part of the text, positive and negative IOD events are not clearly stated, such as:

Line 34: in "negative IOD and IOD-like events", it should be negative IOD-like events? and in Line 231: it should be negative IOD?

Line 279: "long-lasting" - the extremes could be in shorter time scales if daily data is used. The minimum resolution is one month when using monthly data. I suggest the authors need to be careful with using monthly data to derive these conclusions.

Fig. 3 show the composites of the compound events. However, individual negative IOD events may have diverse impacts in the region. Benthuisen et al. (2018) has shown that the 2016 negative IOD sustained the marine heatwaves in the Indonesian-Australian Basin, initiated by the 2015-16 El Nino.

Line 304: MJO can drive sea level extreme events. In an early studies, MJO can also affect marine heatwave occurrences in the region, through modulation of cloud cover and the Australian summer monsoon winds (Zhang et al. 2016). So MJO may have some effects on the compound events as well. Some discussion on this is necessary.

Cai, W., Cowan, T. and Sullivan, A., 2009. Recent unprecedented skewness towards positive Indian Ocean Dipole occurrences and its impact on Australian rainfall. Geophysical Research Letters, 36(11).

Ogata, T., Xie, S.P., Lan, J. and Zheng, X., 2013. Importance of ocean dynamics for the skewness of the Indian Ocean dipole mode. Journal of Climate, 26(7), pp.2145-2159.

Benthuisen, J.A., Oliver, E.C., Feng, M. and Marshall, A.G., 2018. Extreme marine warming across tropical Australia during austral summer 2015–2016. Journal of Geophysical Research: Oceans, 123(2), pp.1301-1326.

Zhang, N., Feng, M., Hendon, H.H., Hobday, A.J. and Zinke, J., 2017. Opposite polarities of ENSO drive distinct patterns of coral bleaching potentials in the southeast Indian Ocean. Scientific reports, 7(1), pp.1-10.

Responses to Reviewers' Comments and Editorial Suggestions on Nature Communications manuscript NCOMMS-21-26867: "Climate-driven sea level extremes and compounding effects of marine heatwaves in coastal Indonesia"
by Han and coauthors.

To meet the 75 characters limit with space, we shorten the title to "*Sea level extremes and compounding marine heatwaves in coastal Indonesia*". We wish to thank the three reviewers for their constructive comments and helpful suggestions, which lead to improvements of our paper. In the revised version, we have carefully considered all comments and revised the manuscript accordingly.

Below, we provide point-by-point response to each of the three referees' suggestions sequentially, repeating portions of each comment in italics, and our responses are in blue. The editor's suggestion on uncertainty calculation has also been addressed.

We are sorry for taking so long to complete the revision; performing the new ROMS and HYCOM experiments in global ocean basins to replace the old runs over the Indian-Pacific basins to thoroughly address the Reviewers' comments took quite some time, but these experiments help improve the consistency of our discussions.

Reviewer #1

"The manuscript addresses a very important science issue on the sea level extremes and marine heat waves that are lasting for many days. They have provided the evidences for the extreme sea levels and marine heat waves along the coast of Java. They also highlighted the significance of the topic considering the current sinking of Java island."

We are thankful for the reviewer's positive comments and for recognizing the importance of this research.

"However, the authors could not provide any convincing hypothesis or mechanisms causing the sea level extremes and marine heat waves over the study region."

In the original MS, we did not explicitly state our hypothesis as the reviewer pointed out. However, we did provide both observational analysis and model experiments to substantiate the mechanisms that cause the sea level extremes (e.g., the original Figs 2c, 4-5) and that cause marine heatwaves, which co-occur with sea level extremes (e.g., Fig 6). They are referred to as sea level height extreme (HEX) and Compound Height-Heat EXtreme (CHHEX). Our discussions focus on two parts:

- **(a)** Detection and understanding of the increased frequency of occurrence and intensity for the HEX events during the 8yr period from 2010-2017;
- **(b)** Causes for the individual HEX & CHHEX event peaks.

To help clarify these points, in the revised MS we improve our presentation by explicitly stating our hypothesis for **(a)**, revising and reorganizing Figs 2, 4 and 5. Now, Fig 4 is dedicated to understanding **(a)**, and Fig 5 (together with Fig 6) is dedicated to exploring **(b)**. Specifically, for **(a)**, we test the hypothesis that anthropogenic global sea level rise

(SLR) and decadal increase in sea level anomaly (SLA) during 2010-2017 due to decadal climate variability account for the increased frequency and intensity of HEXs. We confirm our hypothesis by analyzing observations, reanalysis data and results from OGCM and coupled model experiments.

Regarding **(b)**, since the situations that cause individual HEX events are more complex (especially after we lower the threshold from 2 standard deviations to 90th percentile; discussed below), it is difficult to provide a succinct hypothesis to cover all cases. Instead, we provide topic sentences. For instance, for the CHHEXs, we state in lines 216-217 “*All six CHHEXs occur during negative IOD years, of which five co-occurred La Niña (the negative phase of ENSO), ...*”, and in lines 224-225 “*The negative IOD and La Niña are associated with similar patterns of surface wind anomalies in tropical Indian Ocean (Fig 6b). Their co-occurrence intensifies the wind anomalies; by interacting with seasonal cycle, they result in CHHEXs.*” For HEXs, please see lines 254-258 and 262-267 of the revised MS.

We have also made the discussions of relevant processes clearer, and we hope the reviewer agrees.

“The use of different models is not properly justified, and the detrending also to be relooked at before making the conclusions. The role of decadal variability is attempted but no clear conclusion is made in the manuscript on that.”

We argued that by using different models with different surface forcing fields, we ensure that the simulated key signals exceed cross-model, cross-forcing field differences. Please also see Reviewer 2’s General Comments (top and last paragraphs) that support this argument.

In responding to the reviewer’s comment, we further clarified this point in the revised version, by adding lines 80-82 “*To test the model dependence of simulated signals, we perform additional experiments using an independent OGCM, the Hybrid Coordinate Ocean Model (HYCOM)*”, and lines 88-90 (as in the original MS) “*The multi-dataset and multi-model approach is intended to identify signals that are robust to cross-dataset and cross-model differences. See the Methods section for more details.*”

The role of decadal variability is to enhance the anthropogenic global SLR during 2010-2017, leading to increased frequency and intensity of HEXs during the 8yr period of 2010-2017. In the revised MS, we explicitly spell out the hypothesis (see response above), and we hope the role of decadal variability is more clearly presented.

“The authors suggest negative Indian Ocean Dipole as the prime reason for the sea level extremes and marine heat waves over the Java coast, but the convincing hypothesis or mechanisms are not shown.”

Figs 4c-4d and Table S1 of the original MS show that the five compound events occur in negative IOD years, which are enhanced by La Niña for some events. The mechanisms are discussed in section “**Individual HEX events: mechanisms (lines 194-205)**” for

causes of extreme sea level for all ten sea level extremes, and section “**C-HHEX versus HEX-alone events (lines 206-252)**” for the mechanisms of warm SST anomalies (SSTA) for the five compound events and weak-to-no SSTA signals for the five HEX-alone events. Processes that cause the SSTAs are quantified by the mixed layer heat budget analysis shown in Fig 6.

In responding to the reviewer’s comments, we substantially revised these sections to improve our presentation. Note that in responding to Reviewer 3’s suggestion of using consistent threshold of the 90th percentile for both SLA and SSTA, five additional HEX events with one more CHHEX event are identified. Now we have a total of fifteen sea level extremes; among which six are compounds and nine are sea level alone events. Therefore, some details have changed, but the oceanic processes that cause the events remain the same (see Figs 5 and 6 of the revised MS).

“The different ocean models used in the study are forced by different flux products, making any conclusions (made out of them) very difficult. Use of closed boundary conditions in one set of models and open boundary conditions in another set of models should also be avoided. Such experiments should be carried out with the same model.”

We thank the reviewer for this suggestion. While we intend to use different models with different surface forcing fields to ensure that the key signals exceed cross-model, cross-forcing field differences, we agree with the reviewer that using the same model to perform the set of experiments to assess forcing and processes makes the discussions more consistent.

In responding to this suggestion, we have performed two new experiments using ROMS: ROMS main run (MR) which is the complete solution, and ROMS wind stress (WSTRESS) run that isolates the effects of surface wind stress forcing. Note that the new ROMS experiments are configured to the global tropics (25S-25N) including the tropical Indian, Pacific and Atlantic Oceans, whereas the old ROMS MR of the original MS is confined only to the Indian Ocean basin. In the revised MS, we replaced the old Indian Ocean ROMS with the new global tropical ROMS for all figures, tables, and discussions. The same mixed radiation-nudging boundary condition is used, as in the original ROMS run. Note that the ROMS WSTRESS run is new, to be consistent with the new ROMS MR as the reviewer suggested. In the revised MS, ROMS is used as the main model, because ROMS MR agrees the best with the observations (see revised MS and discussions below).

To demonstrate the SLA and SSTA signals that we have identified exceed cross-model differences, we also performed a new HYCOM MR for global ocean basins. This avoids the closed boundary conditions used in the Indo-Pacific HYCOM runs in the original MS. Meanwhile, the comparisons between the new ROMS MR and HYCOM MR help confirm the effect of global SLR on Indonesian coast sea level extremes (see Methods for detailed discussions). The consistencies among our new and old model experiments using both ROMS and HYCOM clearly demonstrate that the HEX and CHHEXs exceed cross-model differences. **In the revised MS, we focus primarily on the new set of ROMS and HYCOM experiments.**

“They need to validate the models very carefully before making the conclusions.”

On top of the model validations in the original MS, we have significantly enhanced the validation component in the revised MS by

- (1) adding a new table (Extended Table 1) to quantify the satellite observed sea level trend and uncertainty (which is the editor’s comment), satellite and tide gauge observed sea level variability, and the models’ capability (including the ORAS4 reanalysis data) in capturing the observed trend and sea level variability;
- (2) modifying Fig 2c to quantify the time-evolving magnitudes of HEX events, represented by the 90th percentile of SLA with an 8yr sliding window (revised Fig 2c), and compare with satellite observation for their overlapping period;
- (3) adding Fig S4 to show the successful ROMS simulations of SLA and SSTA for the HEX-alone and CHHEX events by comparing with satellite observations (Fig 3);
- (4) adding a subsection “**Model validation**” in **Methods** to focus on discussing the model validations.

“So overall major revision is mandatory by making a clear hypothesis and proving it instead of just speculating many things. If the authors carefully rewrite the manuscript carefully with more convincing experimental evidences, then the manuscript can become a very useful article.”

We appreciate the reviewer for his/her helpful comments and suggestions. We hope that the revised MS meets your expectation.

Reviewer # 2

General comments:

“This is a very interesting and highly original paper on a new combination of extremes with potentially strong impacts, sea level height extremes and marine heatwaves in the Indian Ocean. The paper is well written and the authors present a thorough analysis, combining a suite of different datasets which help provide plausible hypotheses of the drivers behind those compound events. As such, the paper is an important contribution to the emerging field of compound event research.”

We thank the reviewer for such positive remarks, and we appreciate the reviewer’s enthusiasm toward this research.

“One concern would be that the main analysis focuses on compound events in two variables of which one has a strong positive trend (sea level). This is also used to identify drivers of such events. However, due to the strong trend, the strongest driver is anthropogenic climate change, which potentially confounds more detailed conclusions about other important processes. The authors also present some results on the detrended data and I would propose to focus the entire discussion of compound event drivers more on the detrended data to more clearly separate anthropogenic and non-anthropogenic drivers. For instance, I would be interested in the co-occurrence rate of height extremes and marine heatwaves in the detrended time series and the composites of those events, which should be much more representative of short-term processes. In particular, given the very strong correlation between SSHA and SSTA (Fig 2) I would expect a relatively high compound event rate (much more than the 5 event used in the current version).”

We thank the reviewer for this insightful comment. We agree with the reviewer that the strong SLR trend during the satellite era may confound the identification of the compound events.

Following the reviewer’s suggestion, in the revised MS we focus the entire discussion of compound event drivers on the detrended data. Interestingly, by removing the 1993-2018 trend of satellite SLA and SSTA, the six compound events that we identify remain unchanged and their correlation increases only slightly from 0.73 to 0.76 (compare the black and red curves of Fig 2a with those of Fig S6 of the revised MS), even though more SLAs that are slightly below the threshold during 1993-2009 (Fig 2a) are now above the threshold after detrending (Fig S6a), shown as Fig R1 below.

Note that in responding to Reviewer 3’s suggestion of using consistent threshold of the 90th percentile for both SLA and SSTA (see response to Reviewer 3), five additional sea level extremes with one more compound event are identified. Now we have a total of 15 sea level extremes compared to the original 10. Among the 15 sea level extremes, 6 are compounds and 9 are sea level alone events. Therefore, some details have changed, but the relevant oceanic processes remain unchanged (see Figs 5 and 6 of the revised MS).

In the revised MS, all relevant discussions on the HEX and CHHEX events, including their composite analyses, are based on the 15 HEX events (6 CHHEXs and 9 HEX alone).

Fig 2a Monthly satellite SLA (black) and SSTA (red) near Java tide gauge location from 1993-2018, together with Java tide gauge SLA (blue) for its available period of 2007-2016. The horizontal blue (red) line shows the 90th percentile of SLA (SSTA), the threshold value for sea level extreme (marine heatwave).

Fig S6 a Satellite SLA (black) and SSTA (red) with their linear trends of 1993-2018 removed, together with ROMS SLA with 1993-2017 trend removed (blue). The horizontal black (red) line shows the 90th percentile of SLA (SSTA), the threshold value for sea level extreme (marine heatwave).

Figure R1. Combined Fig 2a and Fig S6a.

“Along these lines, that the compound events only occur in the last few years is somewhat obvious if the underlying variables have such strong trends. In this context it is strange to see this phrased as a major result, e.g. by using the word “striking”.”

The word “Striking” is removed in the revised MS.

As mentioned in our response and Fig R1 above, the six compounds remain unchanged after removing the trends of both SLA and SSTA, with four of the six compounds occurring in the 2010-2017 period. This is somewhat surprising. There are two likely reasons for this result:

- (1) The warming trend of SST is weak near the coasts of Java and Nusa Tenggara for the 1993-2018 period; with and without detrending does not have much effect on the extreme SSTA events (>90th percentile).
- (2) Nine out of the ten HEX events during 2010-2017 remain above the 90th percentile after detrending, and only the Dec 2013 HEX-alone event falls below the threshold.

Therefore, the number of CHHEX events for the 2010-2017 period remain unchanged. Even though the warming rate is slow for the 26yr period from 1993-2018, continued warming in the next few decades or century would have larger impact on the SST and thus on the compound events.

Although more SLAs exceed the threshold from 1993-2009 after detrending such as 1993 & 1995 (Fig R1 above), they are not associated with strong SSTAs (i.e., MHWs). Note that all six CHHEXs are induced by strong wind anomalies associated with negative IODs, with five being enhanced by La Niña (two happened before 2010 and three after 2010). The only CHHEX that is associated with negative IOD-only (no La Niña) occurs in May 2013, when the IOD peaks in May (an early onset and early termination event, as discussed in the MS). The moderate interannual SSTA overlying on the high SST in May brings the total SSTA over the 90th percentile.

Has the co-occurrence of negative IOD and La Niña and/or early onset of negative IOD, together with the CHHEX events, increased in the past century? Will they increase in the future due to climate warming? While satellite sea level and SST data are reliable, the data records are too short to address these issues in this paper. However, this can be an important topic for our future research using large-ensemble CMIP6 experiments.

“As I’m not an expert on the oceanic processes, I can merely judge the statistical analyses, which are sound and reveal interesting insights about the processes behind compound events. In particular the employment of multiple different datasets shows robustness of the results and the usage of composites is a good way to reveal important compound event drivers.”

We thank the reviewer for the positive and insightful comments. ROMS closes its mixed-layer heat budget, and the processes govern SSTA are quantitatively estimated in Fig 6.

Further comments:

“- L28: These numbers strongly depend on how extremes are defined. I would use more

qualitative words here, e.g. “the majority” or something similar”

The reviewer is correct that the numbers strongly depend on the threshold used. In fact, using the 90th percentile of SLA in the revised MS (as Reviewer 3 suggested) – instead of the 2 standard deviation (STD) as used in the original MS – we identify 15 sea level extreme events compared to the 10 events defined by 2 STD. In the revised MS, we use more qualitative words for relevant discussions.

“- Examples in the introduction are on eastern Pacific but study regions is the Indian Ocean. Maybe chose more relevant examples for the study region”.

Thanks for the good suggestion. In the revised MS, we removed the eastern Pacific examples and added available examples for the tropical Southeast Indian Ocean that is most relevant to this study. This also helps to reduce the number of references to meet the requirement of 30-50.

“- L42: Calling extremes on a monthly resolution longer-lasting extremes is a bit of a misnomer. One can call something long-lasting if there are many subsequent extremes in a high-resolution dataset but not if the original temporal resolution is coarse. With the same logic one could call a warm year an extremely long-lasting heatwave, which is obviously nonsense.”

We removed the “longer-lasting” in the revised MS.

“- L112: This is to be expected (and not “striking”) if extremes are analysed in a variable with strong upward trend, as stated higher up”

The word is removed. Note that the rising trend due to anthropogenic SLR alone cannot fully explain the concentration of HEX events on the 8yr period of 2010-2017; anthropogenic SLR combined with decadal increase of sea level during this period account for the increased frequency and magnitude of HEX events (see Fig R3 below, which is Fig S3 of the revised MS). By removing the short-term trend of 1993-2017 we essentially remove most of the anthropogenic SLR plus decadal SLA, since the SLAs obtained in the two manners are similar and highly correlated ($r=0.99$; Fig R2).

Fig R2. ROMS SLA with 1993-2017 trend removed (black), and ROMS SLA by first removing anthropogenic global SLR and then remove 8yr lowpass filtered SLA from 1960-2017 but only the 1993-2017 period is shown (blue), which isolates seasonal-to-interannual anomalies. From Fig S6b of the revised MS.

*Figure R3. Time series of monthly mean sea level anomalies (SLAs) averaged over Java coastal area (106°E-114°E, 7°S-9°S; black). **a**, SLAs from ORAS4 reanalysis with the mean of 1960-2017 removed (black curve), and with linear trend plus decadal variability (8yr lowpass filtered using the Butterworth filter) removed (blue); the horizontal black line shows the 90th percentile of the black line; **b**, Same as **a** except for SLA that removes linear trend but retains decadal variability (red); **c**, Same as **a** but for ROMS model simulation; **d**, Same as **b** but for ROMS model simulation. The linear trend of 1960-2017 is used to represent global sea level rise effect. From Fig S3 of the revised MS.*

“- L194 and below: I would strongly suggest to present and discuss the drivers of extreme events based on the detrended time series. Otherwise the process analysis is confounded by long-term anthropogenic trends”

Done. Please also see our response above.

“- Cyan color in Fig. 2 and other figures not legible, please consider using a different color”
 We spent quite a bit time to select more legible colors and decided to use purple instead

of Cyan in Fig. 2. Efforts have also been made to improve legibility in other figures.

Reviewer # 3

“This study use historical observational data and ocean models to describe and investigate the sea level extremes off the Indonesian coast in the southeast Indian Ocean, as well as their compound events with marine heatwaves. It is found that negative Indian Ocean Dipole like condition is the key driver for the development of the compound events, whereas the sea level extreme only events are more driven by regional wind anomalies. Compound sea level and marine heatwaves events are rarely studied so that this research could contribute to the literatures to inspire new research on compound marine extremes. It is a significant amount of research and manuscript is organized with a good structure. Still, the manuscript will need to address the following comments before it can be accepted for publication.”

We thank the reviewer for the positive comments and for recognizing the novelty of the research.

“The definition of marine heatwaves: Hobday et al. (2016) used daily data to define the marine heatwaves, with a 90th percentile criteria. In this study, the sea level extreme is defined as two standard deviations and marine heatwave is defined as one standard deviation, using monthly data. So there need some consistencies. 90th percentile is equivalent to 1.6 standard deviations for normal distributions.”

In responding to the reviewer’s comment, in the revised MS we use the 90th percentile of SLA - instead of the 2-standard deviation - as the threshold. By lowering the threshold value, we identify five more HEX events with one being CHHEX. Now we have 15 HEXs total compared to 10 in the original MS. Among the 15 sea level extremes, 6 are CHHEXs and 9 are HEX-alone events. Therefore, there are some changes in the discussion details, but the oceanic processes that cause the compound events remain the same (see Figs 5 and 6 of the revised MS). In the revised MS, all relevant discussions on the HEX and CHHEX events, including the composite analyses, are based on the 15 total HEXs, 6 CHHEXs and 9 HEX alone events.

“As noted in various studies, there is a positive skewness in the IOD (Ogata et al. 2013; Cai et al. 2009). So, for the eastern IOD pole SST, there could be a negative skewness. How would this SST skewness affect the selection of the marine heatwave events?”

Yes, there is a negative SSTA skewness in the eastern pole of the IOD, as shown by various studies, such as the reviewer suggested Cai et al. (2009) and Ogata et al. (2013) and others (e.g., Hong et al. 2008), including the recent work by two of our coauthors: S. Kido and T. Tozuka (Nakazato, Kido and Tozuka, 2021, scientific reports). The stronger upwelling and colder SSTA during positive IOD (pIOD) years intensify the seasonal upwelling and cold SSTA along Indonesian coast, preventing the HEX and CHHEX from happening. Indeed, none of the HEX and CHHEX events occur in pIOD years. Note that in this paper, the 90th percentile threshold that defines MHWs are based on the SSTA relative to the 30yr-mean SST from 1989-2018. With this definition, the skewness of IOD will not affect the selection of MHW events. However, under anthropogenic climate change, climate model projections suggest increased number of positive IOD and reduced number of negative IOD events due to a mean state change toward a shallower (deeper)

thermocline in the tropical eastern (western) Indian Ocean (e.g., Cai et al. 2018). Intuitively, this positive skewness might reduce the number of extreme sea level and compound events; however, the amplitude of the IODs is projected to increase (Marathe et al. 2021). The shallower thermocline in the eastern pole of the IOD – with continued anthropogenic sea level rise and surface warming albeit with a slower warming rate near Indonesian coast – will favorably precondition the ocean for stronger HEXs and CHHEXs in coastal Indonesia. It is also possible that the number of HEX and CHHEX events will increase, because with a shallower thermocline, even a relatively weaker westerly/northwesterly wind may significantly reduce upwelling, increase upper-ocean temperature and SLA. Relevant discussions can be found in the last paragraph of the MS.

We thank the reviewer for providing the Benthuysen et al. (2018) and Zhang N. et al. (2017) references. In the revised MS, they are added in the opening (introduction) section that discusses ENSO and IOD impacts on MHWs in the eastern Indian Ocean.

“It is useful to compare marine heatwave selection with daily SST data, which is available from various satellite products. Monthly warming may not reflect the full impacts of marine heatwaves on biology processes.”

Previous studies have used monthly SSTA to study MHWs (see relevant references in Hobday et al. 2016). Using monthly data however will smooth out the intensity of individual MHW events defined by daily data (Hobday et al. 2016), and as the reviewer pointed out, the latter better reflects the full impacts of MHWs on marine ecosystems. The monthly warm events, however, are important for affecting climate and also impact marine species.

Following the reviewer’s suggestion, we compare the MHWs identified by monthly SSTA, defined as $SSTA \geq 90^{\text{th}}$ percentile with at least 1 month separation for two consecutive events, with the MHWs identified by daily SSTA. Following Hobday et al. (2016), we define a MHW event (using daily SSTA) as an anomalously warm event that lasts for at least five days with temperatures warmer than the 90th percentile based on the 30yr historical baseline period of 1989-2018, the same baseline period as for the MHWs defined by monthly SSTA. Fig R4 below shows the comparison. We find that most MHWs identified by monthly SSTA, especially those stronger and longer-lasting ones, correspond to a series of MHWs defined by daily SSTA. This figure is added to the revised MS as Fig S12, and the definition and relevant discussions can be found in Methods section.

Figure R4. Time series of monthly and daily SSTA relative to the same 30-year mean of daily SST from 1989–2018, and marine heatwaves (MHWs) defined by both the monthly and daily SSTA. **a**, Monthly (black) and daily (red) time series of SSTA averaged over Java coastal area of (106°E–114°E, 7°S–9°S), the same region as in Figs S2 and S3; the horizontal dotted red line represents the 90th percentile of daily SSTA. **b**, Time series of monthly SSTA (black; same as that in **a**) with the SSTA values exceeding the 90th percentile defined as MHWs, and the MHWs identified from daily data (red), which are defined as discrete prolonged anomalously warm water events when daily SSTAs exceed the 90th percentile and persist for at least 5 days. Note that the monthly SST data are on 1°x1° grid points and daily SST data are on 0.25°x0.25° grid points. From Fig S12 of the revised MS.

“In some part of the text, positive and negative IOD events are not clearly stated, such as: Line 34: in “negative IOD and IOD-like events”, it should be negative IOD-like events? and in Line 231: it should be negative IOD?”

Only the negative IOD that peaks in May 2013 was referred to as negative IOD-like event in the original MS. For this event, the IOD index is negative from April–October, peaks in May and becomes positive in November. So, it is an early onset and early termination negative IOD event. In the revised MS, we just refer it to as a negative IOD event to avoid confusion, since no two IODs are exactly the same.

“Line 279: “long-lasting” - the extremes could be in shorter time scales if daily data is used. The minimum resolution is one month when using monthly data. I suggest the authors need to be careful with using monthly data to derive these conclusions.”

Thanks for this suggestion. We removed “long-lasting” in the revised MS. Please also see our response to Reviewer 2.

“Fig. 3 show the composites of the compound events. However, individual negative IOD events may have diverse impacts in the region. Benthuisen et al. (2018) has shown that the 2016 negative IOD sustained the marine heatwaves in the Indonesian-Australian Basin, initiated by the 2015–16 El Niño.”

Benthuisen et al. (2018) paper is referenced, and relevant discussions are added in the revised MS (lines 46-51).

“Line 304: MJO can drive sea level extreme events. In an early studies, MJO can also affect marine heatwave occurrences in the region, through modulation of cloud cover and the Australian summer monsoon winds (Zhang et al. 2016). So MJO may have some effects on the compound events as well. Some discussion on this is necessary.”

The MJO effects were discussed in the original MS because the monthly mean SLA and SSTA might *alias* some MJO impact signals at periods longer than 60days (>2months). Since monthly mean data exclude most, if not all, MJO signals which peak in 30-60day periods, discussing the effects of atmospheric intraseasonal oscillations (ISOs) is somewhat extraneous and distracting. Therefore, in the revised MS we removed the discussions on ISOs. How and to what extent atmospheric ISOs affect the HEX and CHHEX events in coastal Indonesia deserves a separate, in-depth study. We are currently working along these lines and the Zhang et al. 2016 paper will be referenced there.

REVIEWERS' COMMENTS

Reviewer #1 (Remarks to the Author):

Climate-driven sea level extremes and compounding effects of marine heatwaves in coastal Indonesia by Han et al.

The manuscript addresses a very important science issue on the sea level extremes and marine heat waves that are lasting for many days.

They have provided the evidences for the extreme sea levels and marine heat waves along the coast of Java.

They also highlighted the significance of the topic considering the current sinking of Java island. In the revised manuscript, they have provided the hypothesis or mechanisms causing the sea level extremes and marine heat waves over the study region. The use of different models is also justified, and additional systematic experiments with regional ocean model conducted to strengthen their hypothesis. The role of decadal variability and anthropogenic forcing on the extreme sea levels is also highlighted.

The authors suggest negative Indian Ocean Dipole as the prime reason for the sea level extremes and marine heat waves over the Java coast.

The revised manuscript mostly addressed my concerns and the concerns of the other reviewers, and so I recommend acceptance of the manuscript for publication. I congratulate the authors for the additional model experiments and analysis carried out to substantiate their findings.

Reviewer #2 (Remarks to the Author):

The authors have addressed all my concerns and I have no further comments and recommend publication of this manuscript.

Reviewer #3 (Remarks to the Author):

The authors have addressed most of my previous comments, so I would suggest the manuscript can be accepted for publication after minor revisions. Here are some detailed comments:

The different drivers of the extremes need to be better stated in the Abstract. Both the HEX and CHHEX events are associated with northwesterly wind anomalies. Please provide a summary that the CHHEX events only can occur in a particular season, or under influence of a certain wind pattern/fetch?

Line 293-296, explain why a shallow thermocline would precondition the ocean for stronger HEXs and CHHEXs in coastal Indonesia.

Fig. 1b: add a legend for the wind trend.

Responses to Reviewers' Comments and Editorial Suggestions on Nature Communications manuscript NCOMMS-21-26867A: "Climate-driven sea level extremes and compounding effects of marine heatwaves in coastal Indonesia"
by Han and coauthors.

We thank the three reviewers for taking their time to read our revised MS. While Reviewer 1 and 2 are satisfied with the revised version, Reviewer 3 has a few additional clarification and editorial comments, which have been carefully addressed in the MS.

Reviewer #1

"The manuscript addresses a very important science issue on the sea level extremes and marine heat waves that are lasting for many days....so I recommend acceptance of the manuscript for publication. I congratulate the authors for the additional model experiments and analysis carried out to substantiate their findings."

We are thankful for the reviewer's time and help for improving our MS.

Reviewer # 2

"The authors have addressed all my concerns and I have no further comments and recommend publication of this manuscript."

We are grateful for the reviewer's time and help for improving our MS.

Reviewer # 3

"The authors have addressed most of my previous comments, so I would suggest the manuscript can be accepted for publication after minor revisions."

Here are some detailed comments: The different drivers of the extremes need to be better stated in the Abstract. Both the HEX and CHHEX events are associated with northwesterly wind anomalies. Please provide a summary that the CHHEX events only can occur in a particular season, or under influence of a certain wind pattern/fetch?"

We thank the reviewer for the additional comments. We have modified the Abstract to reflect the reviewer's points (Meanwhile, we have shortened the Abstract to 150 words):

"Both HEXs and CHHEXs are driven by equatorial westerly and longshore northwesterly wind anomalies. For most HEXs, which occur during December-March, downwelling favorable northwest monsoon winds are enhanced but enhanced vertical mixing limits surface warming. For most CHHEXs, wind anomalies associated with a negative Indian Ocean Dipole (IOD) and co-occurring La Niña weaken the southeasterlies and cooling from coastal upwelling during May-June and November-December."

Line 293-296, explain why a shallow thermocline would precondition the ocean for stronger HEXs and CHHEXs in coastal Indonesia.

Done. See the last paragraph of the revised MS, which is copied below:

"The shallower thermocline in the eastern pole of the IOD ... makes the upper-ocean temperature more sensitive to wind-induced Ekman convergence and thus favorably preconditions the ocean for stronger HEXs and CHHEXs in coastal Indonesia."

"Fig. 1b: add a legend for the wind trend."

Done. Please see revised Fig. 1 below.